# A neutral cyclic aluminium (I) trimer

Imogen Squire [1,3], Matthew de Vere-Tucker [1,3], Michelangelo Tritto [1,3], Lygia Silva de Moraes[1], Tobias Krämer [2] ✉ & Clare Bakewell [1] ✉

As part of the quest to develop metal-based redox chemistry beyond the d-block, low oxidation state aluminium complexes have gained wide recognition as discrete and versatile 2-electron reductants. Despite reports of monomeric, dimeric and tetrameric neutral structures, as well as a range of charged aluminyl compounds, neutral trimeric structures have remained notably absent. Furthermore, trimeric nuclearity has previously not been considered when investigating reaction mechanisms. Here, we report two neutral $Al^I$ trimers, *cyclotrialumanes*. The molecules are extensively characterised using both experimental and computational techniques, with the Al−Al bonds described as principally covalent in nature and the trimeric structure shown to be retained in solution. The cyclotrialumanes are highly reactive, activating a range of small molecules and unsaturated substrates (e.g. $H_2$, alkyne, benzene). Most significantly, through a series of extraordinary reactions with ethylene, the cyclotrialumanes are shown to react directly 'as trimers', forming 5- and 7-membered Al−C ring systems.

Over the last 20 years, much emphasis has been placed on the development of 'transition metal-like' reactivity using main group elements[1,2]. Beyond chemical curiosity and an advancement of our fundamental understanding of main group element reactivity, there is a genuine need to find viable, more sustainable alternatives to the platinum group metals. For many years they have been the cornerstone of chemical synthesis, but precious metals are becoming increasingly difficult to mine, with geographical location often making extraction and refinement challenging[3–5]. Aluminium is the most abundant metal in the Earth's crust, and whilst extraction remains an energy intensive process, it is cheap and readily available[6]. It is therefore no surprise that over recent years there has been an eruption of research around this run-of-the-mill element, much of which has focused on Al in the less common +1 and +2 oxidation states[7–10].

Though the first $Al^I$ complex was reported in 1991, the tetrameric $[AlCp^*]_4$ (**I**)[11], there are still relatively few examples of stable $Al^I$ species (Fig. 1). In the realm of neutral molecules, Roesky's β-diketiminate stabilised alumene (**II**) represented the first monomeric $Al^I$ compound and subsequently found great utility as a versatile 2-electron reductant towards a plethora of chemical bonds, owing to its discrete structure and solubility[7,12]. In 2017, the first dialumene, **III**, was reported along with a range of [2 + 2] cycloaddition reactivity[13]. A handful of monomeric and dimeric molecules have followed these seminal reports[14–19], along with examples of aluminyl anions (**IV**), which have proven to be powerful reductants and a rare source of nucleophilic Al[20–22]. Higher order clusters and oligomers, for example 'AlX' fragments where X=halide, are also known and have been isolated in certain cases with judicious choice of donor solvent e.g. the cyclobutane analogue $[Al_4Br_4(NEt_3)_4]$[23,24]. Neutral trimeric univalent $Al^I$ species are, however, notably absent from this list.

Collectively, these examples highlight the flexible electron configuration of Al, which could conceivably mediate redox catalysis. Examples of both facile oxidative addition and reductive elimination occurring at a main group centre are relatively rare but important reactions that allow chemists to develop their fundamental understanding of how electronic configuration might be controlled. We have recently been able to harness the subtle redox reactivity between $Al^I$ and $Al^{III}$ centres; by manipulation of reaction conditions, it is possible to influence a complex equilibrium network and control product formation. These products include dialumene-benzene adducts, which occur through reaction of a transient $Al^I$ intermediate and reaction solvent[25,26]. Whilst not observed experimentally, this result led us to

[1]Department of Chemistry, King's College London, Britannia House, London, UK. [2]School of Chemistry, Trinity College Dublin, The University of Dublin, Dublin 2, Ireland. [3]These authors contributed equally: Imogen Squire, Matthew de Vere-Tucker, Michelangelo Tritto. ✉e-mail: kraemert@tcd.ie; clare.bakewell@kcl.ac.uk

question the possibility of isolating the Al$^I$ species invoked in our mechanism.

Herein, we present the isolation, characterisation and reactivity of a neutral trimeric Al$^I$ complex. The cyclotrialumane, of which we include two examples, is highly reactive and has been shown to react directly as a trimer leading to unique molecular structures.

## Results and discussion

The reduction of aluminium (III) diiodide precursors, **1$^{p\text{-tol}}$** or **1$^{m\text{-xyl}}$**, with four equivalents of finely divided potassium in hexane at room

temperature led to the formation of a deep red-black solution after 16 and 3 h, respectively (Supplementary Fig. S2). In each case, continued stirring led to the complete formation of a single new product (28 and 16 h), which were isolated as dark red crystalline solids (Fig. 2, **2$^{p\text{-tol}}$** 30%; **2$^{m\text{-xyl}}$** 82%).

Crystals suitable for single crystal X-ray diffraction (SCXRD) revealed formation of two cyclic Al$^I$ trimers (Fig. 3). **2$^{p\text{-tol}}$** ($P\bar{1}$) crystallised with two independent molecules in the asymmetric unit (Z' = 2), whereas **2$^{m\text{-xyl}}$** ($P2_1/n$) crystallised as a toluene solvate with Z' = 1. The solid-state structures of both **2$^{p\text{-tol}}$** and **2$^{m\text{-xyl}}$** exhibit planarity across the Al$_3$ core which approximately bisects the ligand backbones (Supplementary Fig. S3). The Al–Al bonds in all three molecules are not equal: this manifests as one significantly shorter and two significantly longer bonds (Fig. 3). The shortest bond length for **2$^{m\text{-xyl}}$** (2.6184(8) Å) is notably shorter than the shortest bond length of the **2$^{p\text{-tol}}$** structures ((2.6308(11) and 2.6553(11) Å for **2$^{p\text{-tol}}$-A** and **2$^{p\text{-tol}}$-B** respectively), with the same pattern observed in the longer bonds. The Al–Al bonds are somewhat longer than the sum of the covalent radii (2.4 Å), however, all are in the range for previously reported Al–Al single bonds (2.5-2.8 Å), and significantly shorter than in the Al$^I$ tetramer, **I** (-2.77 Å)[11,27,28]. The 4-membered rings, formed between the NCN ligand backbone and Al, are not orthogonal to the Al$_3$ core; instead, the ligands are tilted forming a concentric pinwheel structure (Supplementary Fig. S4). All Al–N distances in **2$^{p\text{-tol}}$-A** are longer than in **1$^{p\text{-tol}}$**, consistent with increased electron density at the Al centre.

Whilst monomeric, dimeric and tetrameric Al$^I$ species are known, **2$^{p\text{-tol}}$** and **2$^{m\text{-xyl}}$** are examples of neutral cyclic Al$^I$ trimers, cyclotrialumanes. There are, however, examples of charged, mixed-valence and linear aluminium trimers[19,29–33], and related group 13 structures[34–36]. Trimeric structures are more prevalent in group 14, with heavier analogues of cyclopropane reported for silicon through to lead[37–40]. Most pertinent to this work, is the [Al$_3$]$^{2-}$ anion (and analogous [Ga$_3$]$^{2-}$)[19,30,34]. This aromatic species has significantly shorter Al–Al bond lengths (2.556 Å average), with sodium cations sitting above and below the Al$_3$ core. A neutral thallium trimer bearing a bulky terphenyl ligand has also been reported (Tl–Tl 3.21–3.38 Å), but is proposed to be monomeric in solution[41].

It is notable that reduction of **1$^{p\text{-tol}}$** in benzene or toluene solvent led to formation of the previously reported dialumene benzene

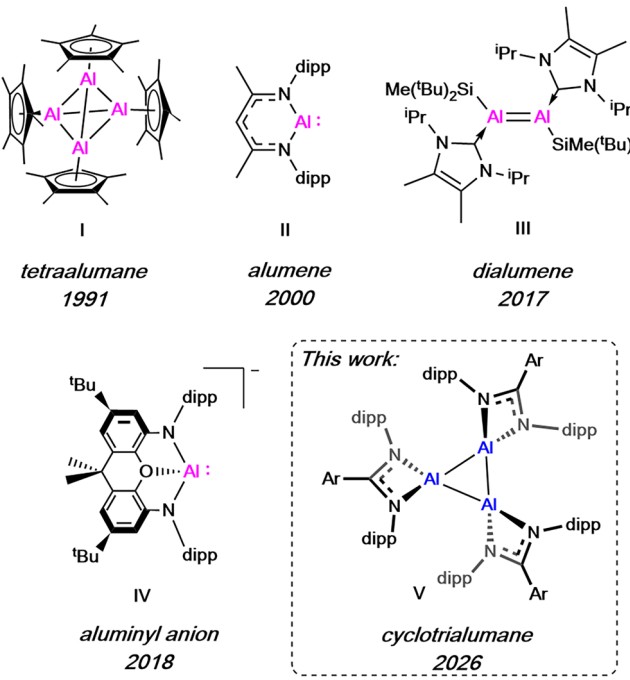

**Fig. 1 | Examples of landmark Al$^I$ complexes. I** Tetraalumane, Schnöckel, 1991; **II** Alumene, Roesky, 2000; **III** Dialumene, Inoue, 2017; **IV** Aluminyl anion, Aldridge and Goicoechea, 2018; **V** This work: Cyclotrialumane.

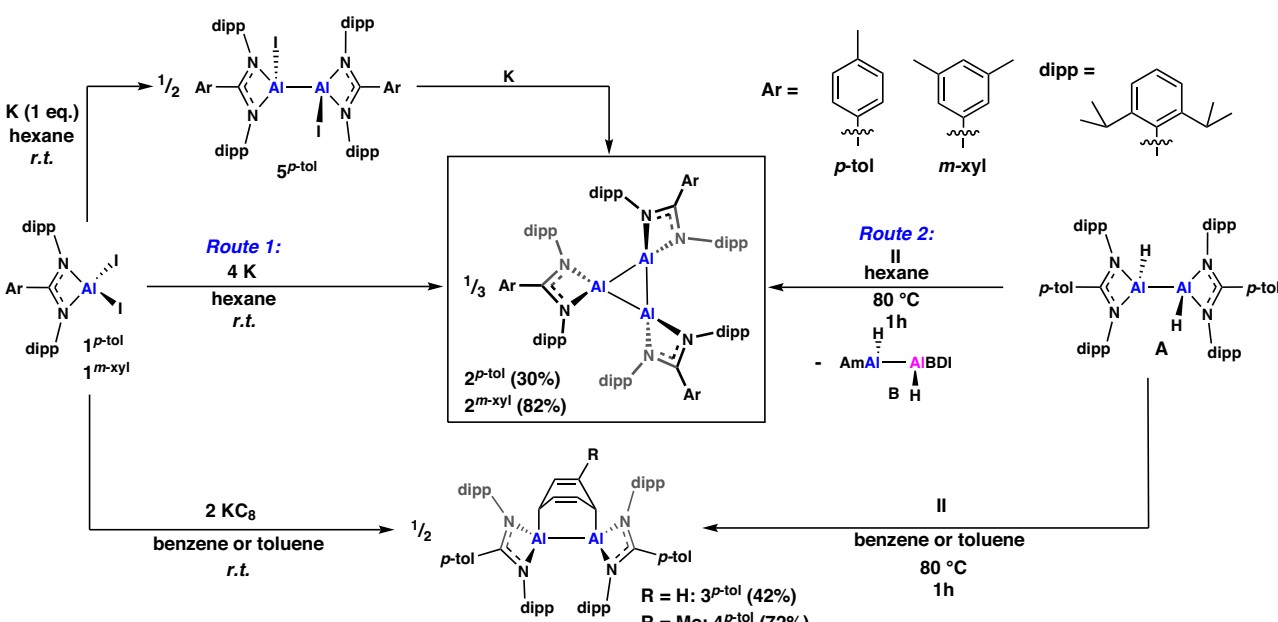

**Fig. 2 | The formation of cyclotrialumanes.** Synthetic routes to the formation of compounds **2$^{p\text{-tol}}$** and **2$^{m\text{-xyl}}$** (via routes 1 and 2) and the related products **3$^{p\text{-tol}}$**, **4$^{p\text{-tol}}$** and **5$^{p\text{-tol}}$**. Reaction conditions, times and yields shown in the figure.

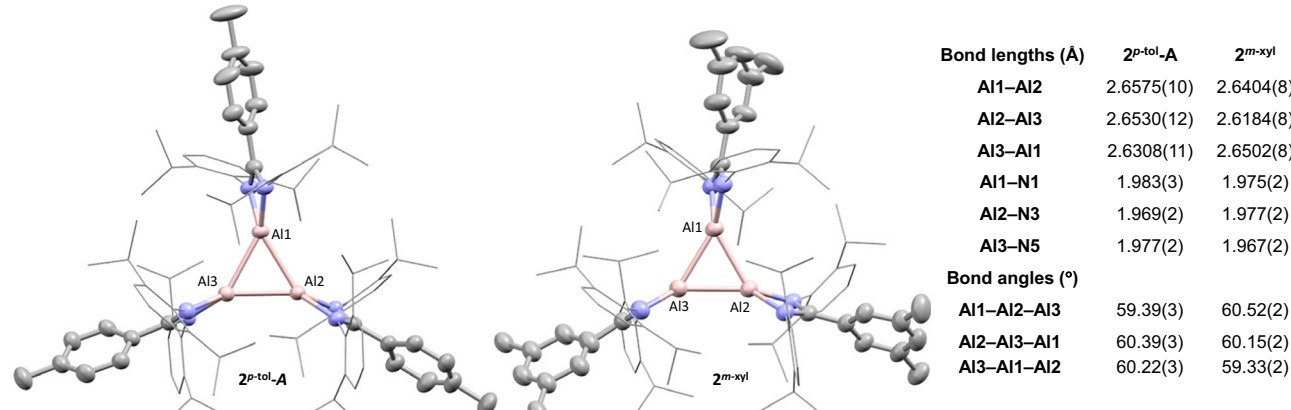

| Bond lengths (Å) | $2^{p\text{-tol}}$-A | $2^{m\text{-xyl}}$ |
|---|---|---|
| Al1–Al2 | 2.6575(10) | 2.6404(8) |
| Al2–Al3 | 2.6530(12) | 2.6184(8) |
| Al3–Al1 | 2.6308(11) | 2.6502(8) |
| Al1–N1 | 1.983(3) | 1.975(2) |
| Al2–N3 | 1.969(2) | 1.977(2) |
| Al3–N5 | 1.977(2) | 1.967(2) |
| Bond angles (°) | | |
| Al1–Al2–Al3 | 59.39(3) | 60.52(2) |
| Al2–Al3–Al1 | 60.39(3) | 60.15(2) |
| Al3–Al1–Al2 | 60.22(3) | 59.33(2) |

**Fig. 3 | Solid state structures of $2^{p\text{-tol}}$-A and $2^{m\text{-xyl}}$.** Selected bond lengths and bond angles are provided in the table. Hydrogen atoms, second molecule $2^{p\text{-tol}}$-B and disorder omitted for clarity, key atoms shown as thermal ellipsoids at 50% probability.

adducts, $3^{p\text{-tol}}$ and $4^{p\text{-tol}}$ (Fig. 2, Supplementary Section 2), however $2^{p\text{-tol}}$ and $2^{m\text{-xyl}}$ are stable in aromatic solvents at room temperature[26]. In benzene-$d_6$, both species have characteristic doublets, corresponding to the $CH_3$ groups of the diisopropylphenyl ligand substituent, two of which lie notably upfield (0.3-0.5 ppm), compared to $1^{p\text{-tol}}$ or $1^{m\text{-xyl}}$, although similar upfield signals are observed for $3^{p\text{-tol}}$ and $4^{p\text{-tol}}$. $2^{p\text{-tol}}$ has two heptets corresponding the methine protons of the isopropyl groups resonating at 3.33 and 4.01 ppm ($2^{m\text{-xyl}}$: 3.33 and 4.00 ppm), as well as a singlet for the backbone $p$-tolyl group at 1.66 ppm. For $2^{m\text{-xyl}}$, a single resonance, at 1.80 ppm, is observed for the $m$-xylyl backbone $CH_3$ substituents. This splitting pattern is surprisingly symmetric, as whilst both $2^{p\text{-tol}}$ and $2^{m\text{-xyl}}$ have $^1$H NMR signals consistent with asymmetry across the plane of the $Al_3$ core, this appears to be due to restricted rotation of the diisopropylphenyl groups as it does not manifest in the $m$-xylyl backbone of $2^{m\text{-xyl}}$.

Aliquots from the reduction of $1^{p\text{-tol}}$ (or $1^{m\text{-xyl}}$), revealed formation of an intermediate species, presumed to be the singly reduced diiododialane. This product could be formed selectively, by treatment of $1^{p\text{-tol}}$ with one equivalent of potassium in hexane (Fig. 2 and Supplementary Fig. S5); $5^{p\text{-tol}}$ forms as pale-yellow crystals which were characterised by SCXRD (Supplementary Fig. S33), although insolubility prevented bulk isolation. This supports $Al_3$ formation being a stepwise process, with relative solubilities of the intermediate likely responsible for differing rates of formation for $2^{p\text{-tol}}$ versus $2^{m\text{-xyl}}$. Upon formation of $2^{p\text{-tol}}$ (and $2^{m\text{-xyl}}$), trace amounts of the dihydrodialane, **A**, were sometimes observed (Supplementary Fig. S7). The exact origin of **A** is unclear but could be a result of small amounts of inter and/or intramolecular C–H activation, mediated by the highly reactive Al centres, leading to Al–H species[19,42].

Samples containing traces of unreacted diiodo- or dihydrodialane proved useful in determining if the trimeric structure was maintained in solution. In each case, DOSY NMR analysis revealed two clear sets of signals, with the diffusion coefficient of $2^{p\text{-tol}}$ (and $2^{m\text{-xyl}}$) consistent with a compound of a significantly higher molecular weight (Supplementary Figs. S8–S10). Furthermore, variable temperature $^1$H NMR (cyclohexane-$d_{12}$) showed no change in the solution structure over the temperature range investigated (10–80 °C, Supplementary Fig. S11). This data is consistent with $2^{p\text{-tol}}$ maintaining its trimeric structure in solution at room temperature, in contrast to Power's thallium trimer[41], as well as at high temperature (at least on the NMR timescale, vide infra). In comparison, a recently reported dispersion stabilised cyclotristannane saw fragmentation above 60 °C[43].

Compound $2^{p\text{-tol}}$ can also be formed through reaction of dihydrodialane **A**, and Al$^I$ precursor **II**. Heating equimolar quantities of **A** and **II** in cyclohexane at 80 °C led to the formation of a dark red-black

solution after 30 min. NMR analysis revealed the presence of $2^{p\text{-tol}}$, alongside the asymmetric dihydrodialane (**B**) and unreacted starting materials (Supplementary Fig. S12). Manipulation of the stoichiometries and reaction conditions did not alter this distribution, indicating that these four compounds exist in equilibrium. Despite being unable to drive the reaction towards $2^{p\text{-tol}}$, it was possible to fractionally isolate small amounts of crystals from the reaction, but not to isolate the compound with bulk purity. Formation of $2^{p\text{-tol}}$ likely occurs though the disproportionation of **A** in the presence of **II**, forming a transient Al$^I$ monomer, which goes on the trimerise (Supplementary Fig. S13). Although reduction to **2** proceeds using either potassium or **II**, the use of excess reducing agent led only to decomposition, with further reduction not observed under the conditions investigated.

In a further noteworthy observation, the SCXRD data for **2** indicates significant electron density residing in the $Al_3$ plane but outside of the Al–Al bond path (Fig. 4D). Initially, the electron density was modelled as hydride ligands, with a planar $Al_3H_3$ ring comprising of three [LAl·H]$^\cdot$ radical units. The diamagnetic NMR spectra, absence of an electron paramagnetic resonance (EPR) signal and onwards reactivity are, however, inconsistent with a radical species (Supplementary Figs. S14, S15). Known 6-membered aluminium halide species ($Al_3X_3$) and oxo-bridged $Al_3O_3$ compounds have significantly longer Al···Al distances[14,44–46], thus ruling out co-crystallisation with halide or oxo-bridged trimers. This combination of data suggests that the unassigned electron density observed in the crystal structure can be attributed to the Al–Al bonds.

Given the unique structure and unusual electronic features of $2^{p\text{-tol}}$ and $2^{m\text{-xyl}}$, we sought to undertake detailed Density Functional Theory (DFT) analysis using the M06-2X-D3 exchange-correlation functional in combination with def2-TZVP and def2-SVP basis sets placed on Al/N and C/H centres, respectively. All the data reported is for $2^{p\text{-tol}}$, with $2^{m\text{-xyl}}$ giving virtually identical results (Supplementary Section 5). The bond parameters of the DFT-optimised geometry of $2^{p\text{-tol}}$ in its singlet ground state match the values of their crystallographic counterparts closely. The slightly inequivalent Al–Al bond lengths seen experimentally are well reproduced (2.640 Å, 2.635 Å, 2.621 Å) with the pattern of two long and one short bond persisting. This effect can be attributed to the steric impact of the ligands, preventing the system from adopting local $C_3$ symmetry. For a truncated model (Ar groups = phenyl) the $C_3$-symmetric geometry is a local minimum (Al–Al 2.59 Å). TD-DFT calculations gave absorption bands at 330 and 440 nm, mapping well onto the experimentally determined $\lambda_{max}$ (311 and 434 nm, Supplementary Figs. S16, S41). The good performance of M06-2X in describing charge transfer has been previously reported and ascribed to its high amount of HF exchange which leads to reasonably balanced

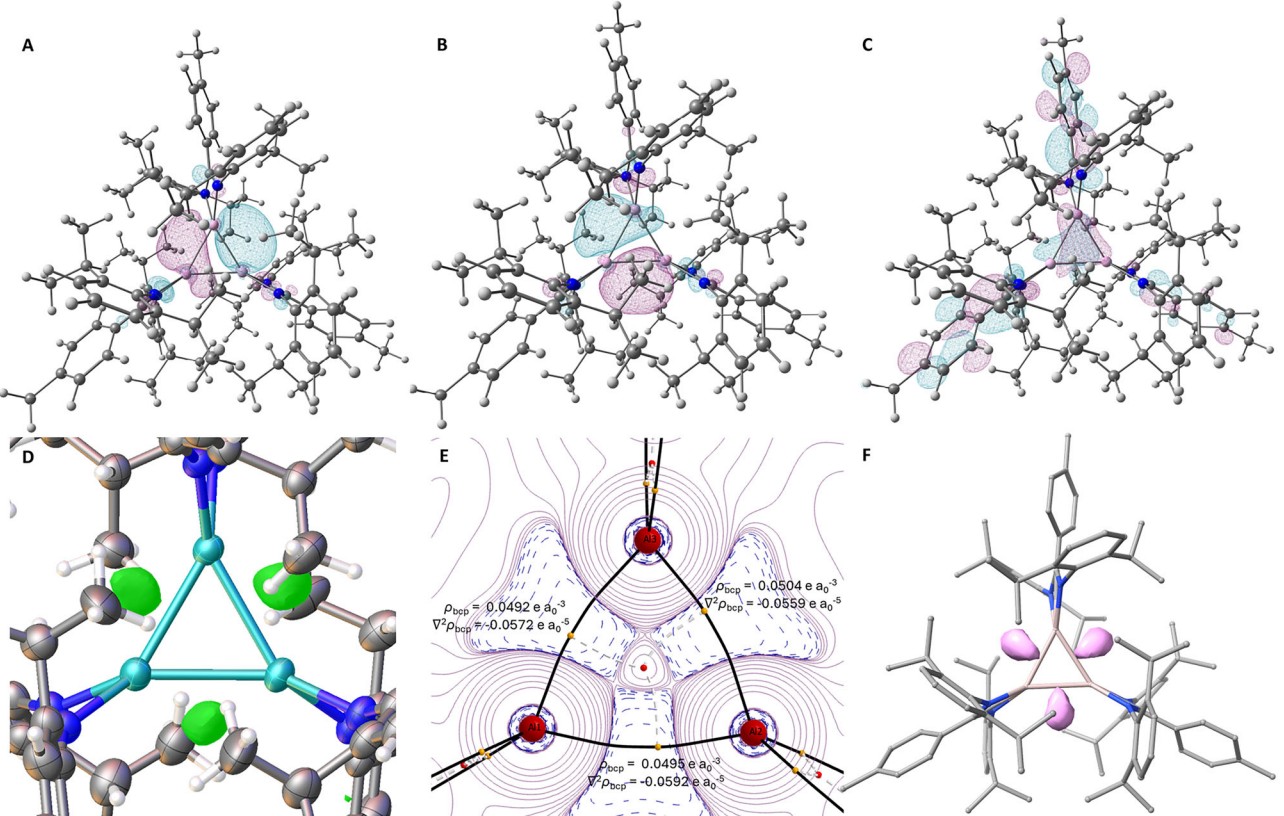

**Fig. 4 | Computational and experimental analysis of 2^p-tol. A** HOMO-1; **B** HOMO; **C** LUMO; **D** Residual electron density plot (green) from SCXRD data; **E** QTAIM topology plot for the (negative) Laplacian of the electron density (-$\nabla^2\rho$) highlighting areas of charge concentration (blue) and depletion (red). The values of electron density ($\rho$) and ($\nabla^2\rho$) are highlighted at each bcp; **F** Isosurface plot for the ELF of 2^p-tol, visually highlighting the V(Al,Al) valence basins adjacent to the Al–Al bonds (pink).

treatment of medium- to long-range correlation[47,48]. The band at 330 nm corresponds to metal-to-ligand charge transfer (MLCT), whereas the band at 440 nm corresponds to frontier orbital transitions principally located on the $Al_3$ core (Supplementary Tables S26, 27). The HOMO and HOMO-1 are two effectively degenerate orbitals (Fig. 4A, B) of tangential spatial character. A radial combination of Al based p-orbitals is also observed at lower orbital energy (HOMO-23) pointing towards the centre of the $Al_3$ ring (Supplementary Figs. S52, S53). Collectively, these orbitals accommodate six valence electrons and form part of the $\sigma$-bonding skeleton of the central $Al_3$ core. The LUMO represents a delocalised bonding $\pi$-symmetric orbital extending above and below the $Al_3$ core (Fig. 4C) and is separated from the HOMO by 4.3 eV.

Analysis of the Lewis structure using Natural Bond Orbitals (NBO) reveals three distinct Al–Al 2-centre-2-electron $\sigma$-bonds of dative nature, resembling the bonding situation recently proposed for a related $Al_6$ cluster[49]. Additionally, dative interactions are present between each ligand and the Al centres aligning with the assigned optimal NBO Lewis structure containing $Al_3^{3+}$ (NPA charges of +0.66e per Al centre) surrounded by anionic ligands. Analogous to cyclopropane, the localised $\sigma$-orbitals of each Al–Al bond are not aligned with the bond vector but protrude towards the outside of the bond axis. Wiberg Bond Indices (WBI) of all Al–Al interactions (~0.97) are indicative of formal single bond character. Additional stabilisation of the cyclotrialumane ring results from hyperconjugative interactions between each Al–Al $\sigma$-bond and the vacant acceptor p-orbitals on the opposite Al centres (Supplementary Fig. S59 and Supplementary Table S37)[50]. This picture of the electronic structure is further supported by Energy Decomposition Analysis (EDA), which also indicates that the present ligand supplies less dispersion energy than, for example, the 2,4,5-

triscyclopentylphenyl ligand used to stabilise the cyclotristannane reported by Power and co-workers (Supplementary Figs. S46, S47)[43].

Quantum Theory of Atoms in Molecules (QTAIM) calculations located three bond critical points (bcp) in the topology of the electron density $\rho$ between each of the Al–Al vectors (Fig. 4E), each slightly displaced concentrically from the geometric bond centre. A ring critical point (rcp) is present at the $Al_3$ ring centre. The values of $\rho_{bcp}$ and its associated Laplacian of $\rho$ ($\nabla^2\rho_{bcp}$) at the Al–Al bcps suggest the bonding character to be of dative nature. The Cremer and Kraka energy density parameters ($H(r) = G(r) + V(r)$) reinforce this assignment, highlighting negative energy density values at all three bcps ($H(r)$: −0.019 to −0.021 a.u.)[51]. The system also reveals significant ring strain, as evidenced by the curved bond paths (bp) between the Al nuclear attractors.

Finally, the Electron Localisation Function (ELF) was interrogated to identify regions of space that can be identified with electron pairs. The topological analysis of the ELF reveals the presence of three valence basins associated with the electron pair density of the Al–Al bonds localised adjacent to the $Al_3$ triangular core. These regions seem to coincide with the residual electron density observed crystallographically, underscoring the picture of the electronic structure obtained from NBO and QTAIM and are consistent with the presence of strained bonds (Fig. 4F)[51–53].

As the cyclotrialumanes can be viewed as a weakly bonded analogue of cyclopropane, their formation could also be viewed as analogous to [1 + 2] pericyclic addition of a singlet carbene to an alkene. In the case of **2**, the cyclotrialumane likely forms through the [1 + 2] addition of an alumene to a dialumene, both of which have already been proposed as transient, highly reactive intermediates in related work[25,26]. Comparing the relative Gibbs free energies of **2** versus three

**Fig. 5 | The reactivity of cyclotrialumanes.** The reaction of **2^p-tol** with benzene (**3^p-tol**), trimethylsilyl acetylene (**6^p-tol**), methyl iodide (**7^p-tol**) and dihydrogen (**8^p-tol**) and the reaction of **2^m-xyl** with benzene (**3^m-xyl**), trimethylsilyl acetylene (**6^m-xyl**) and dihydrogen (**8^m-xyl**). Reaction conditions, times and yields shown in the figure.

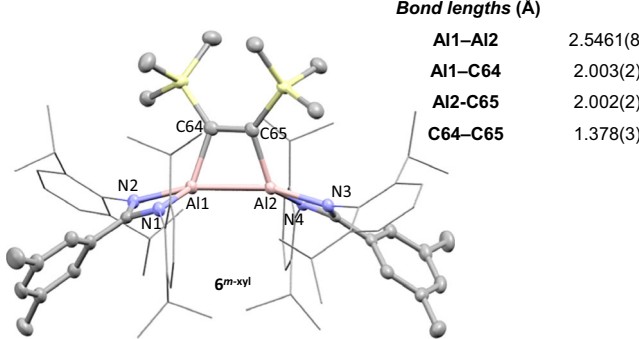

| Bond lengths (Å) | |
|---|---|
| **Al1–Al2** | 2.5461(8) |
| **Al1–C64** | 2.003(2) |
| **Al2-C65** | 2.002(2) |
| **C64–C65** | 1.378(3) |

**Fig. 6 | Solid state structure of 6^m-xyl.** Selected bond lengths and bond angles are provided in the table. Hydrogen atoms omitted for clarity, key atoms shown as thermal ellipsoids at 50% probability.

alumene (−38.7 kcal mol⁻¹) and 1.5 dialumene (−24.5 kcal mol⁻¹) fragments shows trimerisation is highly favourable (Supplementary Fig. S64), with this increased stabilisation attributed primarily to the hyperconjugative interactions. This suggests dissociation of the cyclotrialumane is unfavourable and hints, along with experimental observations, that **2** may react directly as a trimer. To probe the integrity of the cyclotrialumane structure, we set about exploring the reactivity of **2^p-tol** and **2^m-xyl**.

At room temperature, **2^p-tol** and **2^m-xyl** are stable in aromatic solvents for prolonged periods (>1 week). However, when **2^p-tol** or **2^m-xyl** are heated at 80 °C in benzene for 3 h the formation of the previously reported dialumene-benzene adduct, **3^p-tol** (78%) or new compound **3^m-xyl** (75%) was observed (Fig. 5). We have previously proposed that dialumene-arene adducts of this type most likely form via a transient dialumene[26]. Al[I] complexes are also well known to react with internal alkynes, which have been used to probe the nuclearity of the reactive species[18]. Combining **2^p-tol** or **2^m-xyl** with a slight excess of bis(-trimethylsilyl)acetylene (BTMSA) in cyclohexane-$d_{12}$ saw no reaction at room temperature, however, upon heating the reaction (80 °C, 6 h) a loss of colour was observed. In both cases the ¹H NMR spectrum revealed complete conversion to a new product containing broad C$H_3$ resonances, consistent with a sterically crowded coordination environment which resolved to broad multiplets at high temperature (Supplementary Figs S19, S20). This is consistent with formation of **6^p-tol** (81%) and **6^m-xyl** (65%), from a [2 + 2] cycloaddition of an alkyne and a dialumene (Supplementary Fig. S21). SCXRD confirmed the structure of **6^m-xyl**, with an Al–Al bond length of 2.5461(8) Å and a C–C bond length for the BTMSA unit of 1.378(3) Å (Fig. 6). Both are slightly longer than in previously reported [2 + 2] products[54,55]. The lack of reaction at room temperature, coupled with the [2 + 2] and [2 + 4] products

observed from reaction of **2** with benzene and BTMSA, indicates sterically bulky substrates are unable to react directly with the trimeric core. Instead, **2** likely partially dissociates into Al[I] fragments at persistent high temperature. Whilst fragmentation was not observed in the variable temperature NMR data, the strong thermodynamic driving force towards product formation would render transiently formed Al[I] species highly reactive.

In contrast, reaction of the cyclotrialumanes with methyl iodide (MeI) or dihydrogen (H₂) proceeded quickly at room temperature. Treatment of **2^p-tol** with three equivalents of MeI in benzene-$d_6$ at room temperature saw immediate reaction, with NMR analysis revealing clean addition of the Me-I bond across an Al[I] centre (Fig. 5, 70%). **7^p-tol**, which could also be independently synthesised, has a ¹H NMR spectrum consistent with asymmetry across the Al-ligand plane and an Al−C$H_3$ group at 0.23 ppm (3H). Addition of an excess of H₂ gas (-1 bar) to **2^p-tol** or **2^m-xyl** in benzene-$d_6$ also led to a rapid quenching of the dark-red solution. ¹H NMR analysis revealed the formation of the known dihydrodialane, **A** (65%), or the new species **8^m-xyl** (45%) (Supplementary Fig. S22). Initial examination of the ¹H NMR spectra indicate the presence of observable reaction intermediates (Supplementary Fig. S23), though it has not yet been possible to isolate such species. **A** does not further react with H₂, even at elevated temperature; the reaction of **2^p-tol** and **2^m-xyl** with just 0.5 equivalents of H₂ per Al contrasts with some other Al[I] species[19,56].

With both MeI and H₂, facile reactivity impedes the ability to draw conclusions about the nuclearity of the Al[I] species; a substrate of intermediate size was therefore investigated. The addition of ethylene (1 bar) to **2^p-tol** or **2^m-xyl** in benzene-$d_6$ at room temperature saw an instant colour change from dark red to bright orange (Supplementary Fig. S24). Immediate removal of excess gas, followed by ¹H NMR analysis (800 MHz) showed complete consumption of the trimer and formation of a new product consistent with addition of a single molecule of ethylene into the Al₃ core, **9^p-tol** and **9^m-xyl** (Fig. 7, >98%). Characteristic Al-C$H_2$- signals were observed at 0.32 and 1.38 ppm (**2^p-tol**) and 0.31 and 1.41 ppm (**2^m-xyl**), with twelve doublets (-CH(C$H_3$)₂) each integrating to six protons also resolved (Supplementary Figs. S97, S99). SCXRD further confirmed the structure of **9^m-xyl** (Fig. 8), which showed the ethylene fragment to be puckered out of the Al₃ plane. The C−C bond length (1.557(2) Å) is consistent with a single bond and the Al−C bond lengths (1.979(2) and 1.984(2) Å) are relatively long, but in-line with coordination to low oxidation state Al centres. Comparison of Al−N bond lengths showed little variation between the three Al centres (Supplementary Table S17), indicating delocalisation of electron density, despite the formal Al[I] and Al[II] oxidation states. Isolated samples of **9^p-tol** and **9^m-xyl** are stable in solution for prolonged periods of time (>1 week) and do not show evidence of reversibility with application of vacuum or at increased temperature (Supplementary Fig. S26).

However, if **9^p-tol** or **9^m-xyl** are left to stand in an excess of ethylene (-6 equivs) at room temperature further reactivity is observed. New

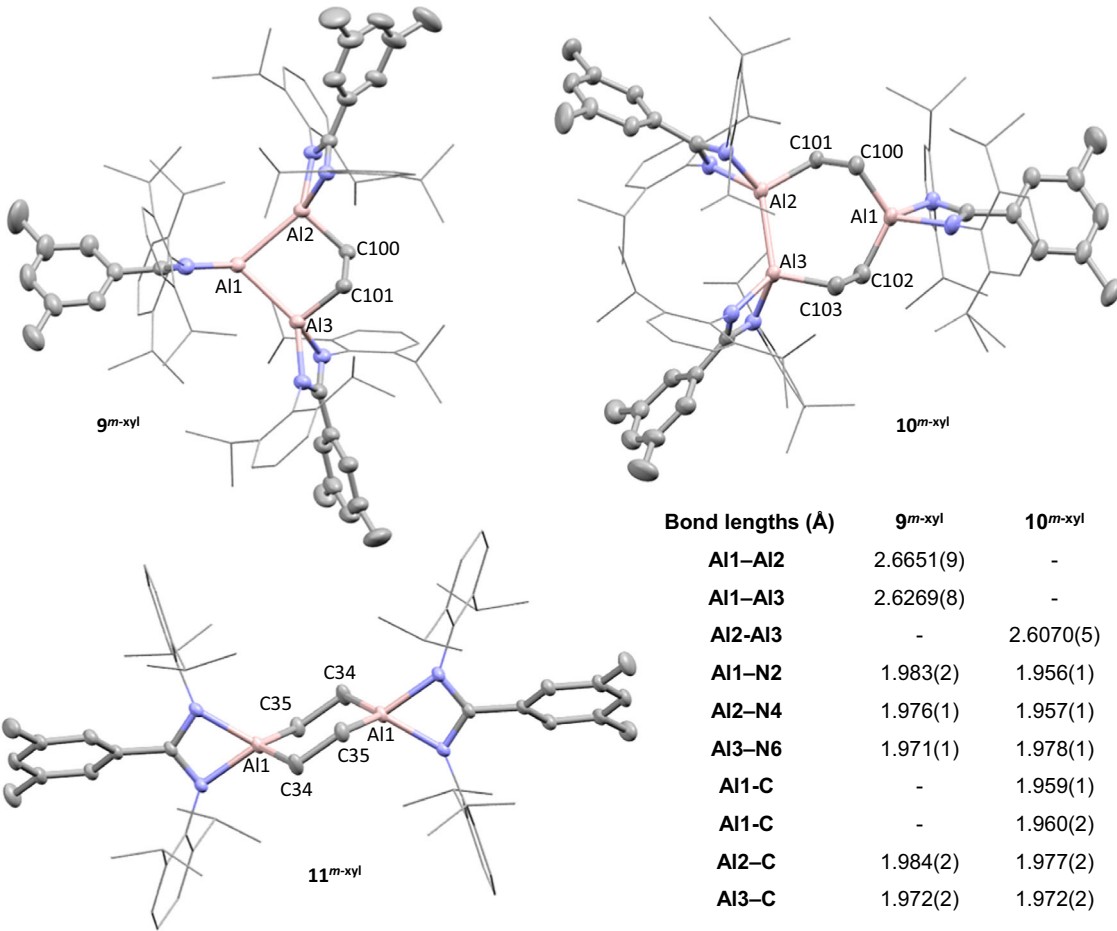

**Fig. 7 | The reactivity of cyclotrialumanes with ethylene.** The reaction of **2**[p-tol] and **2**[m-xyl] with ethylene, which initially forms compounds **9**[p-tol] and **9**[m-xyl]. Continued reaction with excess ethylene leads to formation of **10**[p-tol] and **10**[m-xyl], **11**[p-tol] and **11**[m-xyl], **12**[p-tol] and **12**[m-xyl], with % product formation shown in the table. *Indicates % NMR yield.

**Fig. 8 | Solid state structures of 9**[m-xyl]**, 10**[m-xyl] **and 11**[m-xyl]**.** Solid state structures of products from the reaction of **2**[m-xyl] with ethylene. Selected bond lengths and bond angles are provided in the table. Hydrogen atoms, and disorder omitted for clarity, key atoms shown as thermal ellipsoids at 50% probability.

| Bond lengths (Å) | 9[m-xyl] | 10[m-xyl] |
|---|---|---|
| Al1–Al2 | 2.6651(9) | - |
| Al1–Al3 | 2.6269(8) | - |
| Al2-Al3 | - | 2.6070(5) |
| Al1–N2 | 1.983(2) | 1.956(1) |
| Al2–N4 | 1.976(1) | 1.957(1) |
| Al3–N6 | 1.971(1) | 1.978(1) |
| Al1-C | - | 1.959(1) |
| Al1-C | - | 1.960(2) |
| Al2–C | 1.984(2) | 1.977(2) |
| Al3–C | 1.972(2) | 1.972(2) |

signals appear in the ¹H NMR within 1–3 h (depending on sample concentration), and after ~10 h, all **9**[p-tol] (or **9**[m-xyl]) was consumed and three new species were identified. Through fractional crystallisation it has been possible to isolate and characterise two of these (Fig. 7).

For both **9**[p-tol] and **9**[m-xyl], the major product represents insertion of a second equivalent of ethylene into one of the Al–Al bonds (**10**[p-tol] (45%), **10**[m-xyl] (42%), Scheme 3). SCXRD of **10**[m-xyl] revealed a puckered 7-membered ring (Fig. 8), with C–C bond lengths of 1.556(2) and 1.559(2) Å, similar to **9**[m-xyl]. The remaining Al–Al bond is 2.6070(5) Å, which is significantly shorter than in **2**[m-xyl] but still well within the reported range for Al–Al single bonds. The difference in the Al–N bond lengths and N–Al–N bond angles, are all consistent with Al1 (Al[III])

occupying a higher formal oxidation state than Al2 and Al3 (Al[II]). The Al–C bond lengths are also significantly shorter to Al1, as would be expected for a more electropositive centre.

Whilst only determined to make-up ~30% of the reaction mixture, aluminium ethylene dimers **11**[p-tol] or **11**[m-xyl] were relatively insoluble and could also be cleanly isolated (Fig. 7). The ¹H NMR spectra are highly symmetric, with singlets for the $C_2H_4$ protons (8H) at 1.20 (**11**[p-tol]) and 1.21 ppm (**11**[m-xyl]). SCXRD analysis revealed **11**[m-xyl] crystallised in a chair shaped conformation, whereas **11**[p-tol] adopted a twisted conformation (Fig. 8 and S38). The C–C (1.553(4) Å (**11**[p-tol]); 1.5484(18) Å (**11**[m-xyl])) and Al–N bond lengths (and N–Al–N bond angle) are in line with Al[III] species and consistent with related structures[19,57]. The final product has not

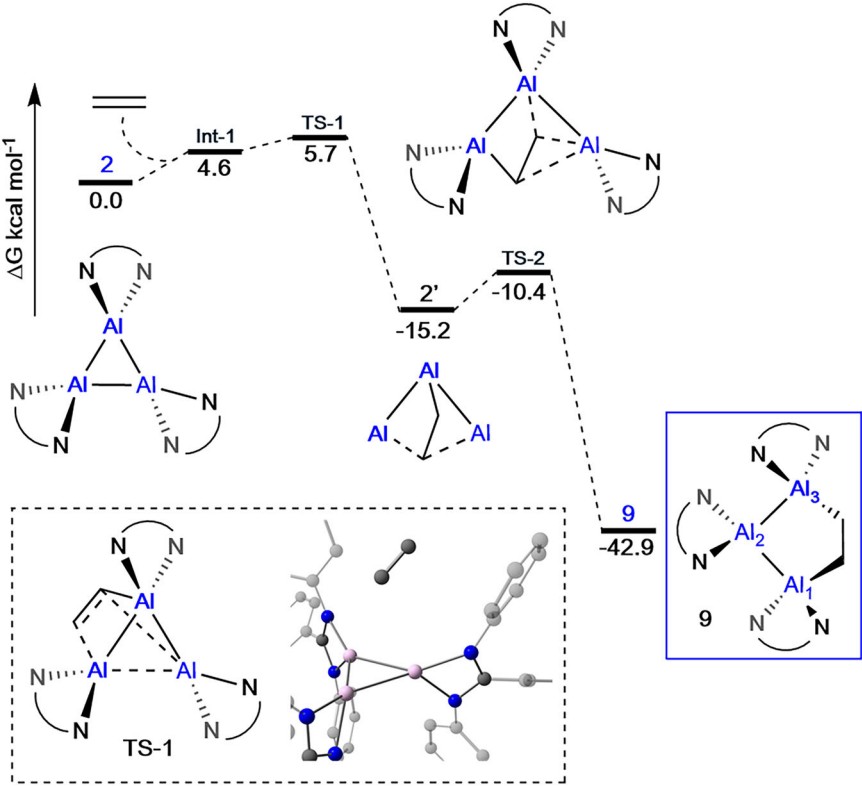

**Fig. 9 | Calculated reaction pathway for the formation of 9.** Calculated reaction pathway for the formation of **9** from **2** and ethylene using a truncated ligand system (Ar$_1$ = Ar$_2$ = Ar$_3$ = Ph). M06-2X-D3/def2-TZVP; single point energies corrected for solvent (SMD, benzene). Gibbs Free Energies (kcal mol$^{-1}$). Isolated reaction products highlighted with blue box. Key transition states are depicted in dashed box. All other intermediates and transition states are found in Supplementary Figs. S67, S68.

been isolated to date, however, analysis of the highly symmetric $^1$H NMR spectrum reveals distinctive signals at 0.55 (t) and 2.05 (pent) ppm (*m*-xyl). This is consistent with species containing a -C$_4$H$_8$- unit, formed through coupling of two ethylene molecules (Supplementary Figs S27, S28) and which has been tentatively assigned as the aluminacyclopentanes, **12$^{p\text{-}tol}$** or **12$^{m\text{-}xyl}$** (Fig. 7). An attempt to independently synthesise the proposed aluminacyclopentane gave data consistent with what is observed experimentally, and with that of a related structure proposed in the literature[19], though it has not been possible to isolate crystals suitable for SCXRD.

The 5- and 7-membered metallocyclic products (**9–10**) are without precedent for both transition and main group metals, and whilst analogues of **11** and **12** have previously been reported, the formation of all four compounds is of fundamental interest. As such, we sought to interrogate the mechanism by which **2** reacts with ethylene. It is noteworthy that **9** forms from **2** instantaneously and irreversibly, therefore products **10–12** originate from the stable intermediate **9**, not from **2**. Furthermore, once formed, the composition of **10–12** (*m*-xyl: 42:32:26) does not change over time and no interconversion is observed upon heating for prolonged periods at 80 °C. This suggests all three compounds form via reaction of **9** with ethylene, and there are likely multiple competing pathways in play. Kinetic analysis of the reaction of **9$^{m\text{-}xyl}$** with ethylene shows **10$^{m\text{-}xyl}$**-**12$^{m\text{-}xyl}$** all form simultaneously, though at differing rates (Supplementary Fig. S29). When a larger excess of ethylene is used (~2000 equivs), products **10$^{m\text{-}xyl}$**-**12$^{m\text{-}xyl}$** form significantly faster (15 min), but the product distribution remains the same. However, the product distribution changes when **9$^{m\text{-}xyl}$** and excess ethylene react at higher temperature (80 °C: **10$^{m\text{-}xyl}$** 27%; **11$^{m\text{-}xyl}$** 28%; **12$^{m\text{-}xyl}$** 45%). This suggests that **10$^{m\text{-}xyl}$** is the kinetic product, whilst **12$^{m\text{-}xyl}$** is a thermodynamic product.

To gain further insight into the formation of **9** and its further reaction with ethylene, the mechanism was probed by DFT. Due to the size of **2$^{p\text{-}tol}$/2$^{m\text{-}xyl}$**, the truncated ligand system was again employed (Ar groups = phenyl). The truncated trimer **2**, reacts with a molecule of ethylene via a low energy transition state (**TS-1**, 5.7 kcal mol$^{-1}$), initially forming the Al$_3$-ethylene adduct, **2'**, the formation of which is exergonic (−15.2 kcal mol$^{-1}$, Fig. 9). From **2'**, only a very slight movement of the ethylene fragment is required, to result in the insertion product **9**, via a second low energy transition state (4.8 kcal mol$^{-1}$). Such low energy transition states are consistent with facile room temperature reactivity. Formation of the 5-membered ethylene insertion product, **9**, also has a strong thermodynamic driving force (−42.9 kcal mol$^{-1}$ versus starting materials), as would be expected upon the release of ring strain and is consistent with a lack of reversibility. From the isolatable intermediate **9**, compounds **10, 11** and **12** are formed.

Initially, we attempted to fragment **9** into smaller reactive intermediates (Supplementary Fig. S65) capable of reacting further with ethylene via mechanisms previously proposed in the literature[13,19,58]. However, these pathways were all highly endergonic and inconsistent with a room temperature reaction.

Instead, an alternative lower energy pathway, proceeding via an adduct of **9** and ethylene (**9'**) was located. The adduct, **9'**, features an ethylene unit bridging the two aluminium centres (Al$_1$ and Al$_3$, Fig. 10) involved in bonding to the inserted ethylene molecule. From this intermediate, two different reaction pathways could be identified. The first pathway proceeds via a low energy transition state (**TS-4'**, 3.3 kcal mol$^{-1}$) during which a bridging carbon of **9'** attacks the middle aluminium (Al$_2$), leading to the second ethylene insertion product **10** (via the alternative conformer **10'**). The second pathway sees the bridging adduct formally insert between the two aluminium centres to which it is bound (Al$_1$ and Al$_3$), creating two new Al−C bonds to form **11**

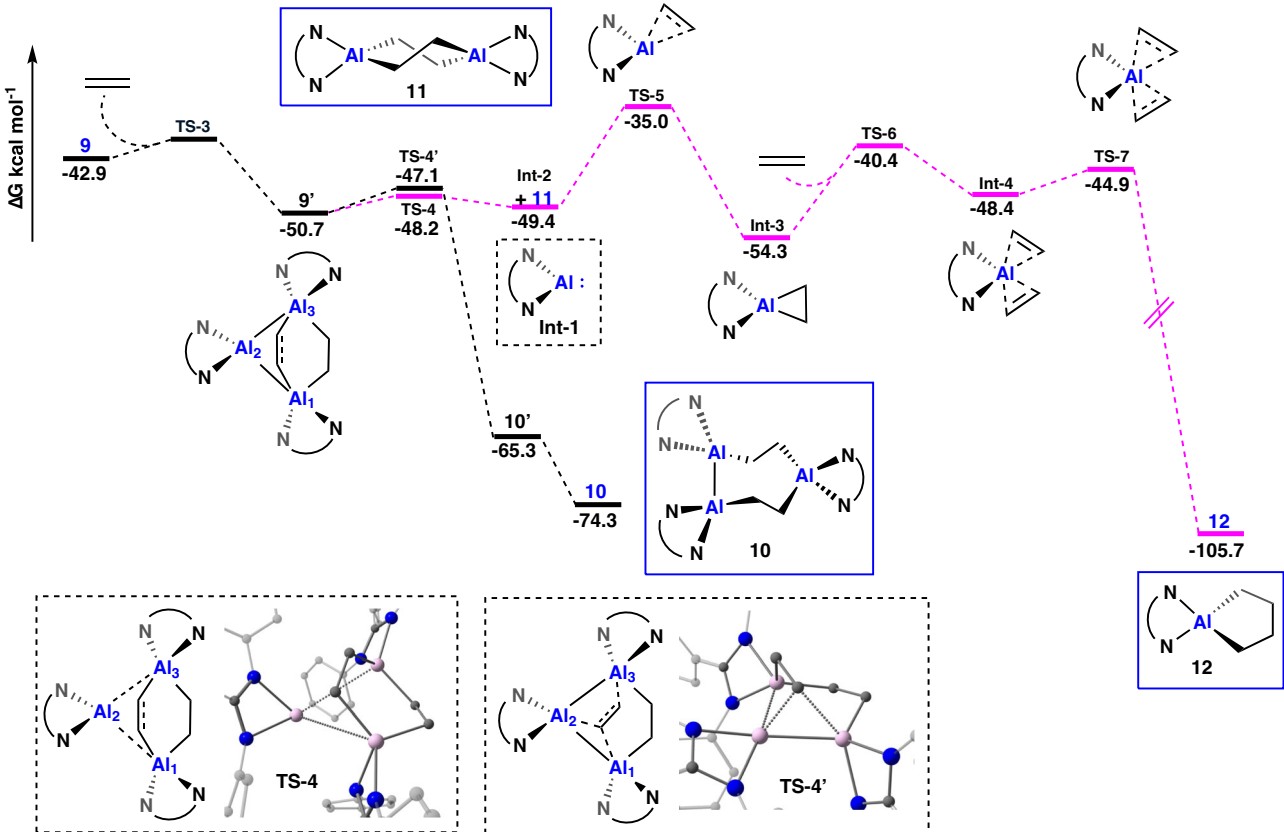

**Fig. 10 | Calculated reaction pathway for the formation of 10, 11 and 12.** Calculated reaction pathway for the reaction of **9** with excess ethylene leading to the formation of **10**, **11** and **12**. Truncated ligand system (Ar₁ = Ar₂ = Ar₃ = Ph); M06-2X-D3/def2-TZVP; single point energies corrected for solvent (SMD, benzene). Gibbs Free Energies (kcal mol⁻¹). Isolated reaction products highlighted with blue box. Key transition states are depicted in dashed box. All other intermediates and transition states are found in Supplementary Figs. S67, S68.

and concomitantly releasing an alumene fragment (**Int-2**). The transition state (**TS-4**) for this pathway is 1.1 kcal mol⁻¹ lower in energy than **TS-4'**, with product formation (**Int-2+11**) slightly endergonic and the relatively high energy of **Int-2** offset by the favourable formation of **11**. **Int-2** can then go onto react with two further equivalents of ethylene in a stepwise fashion with the rate determining step the first addition of ethylene to **Int-2**. Ultimately this leads to **12**, which is stabilised by −105.7 kcal mol⁻¹ relative to the reactants. A related pathway has previously been reported by Power and co-workers to rationalise the formation of an aluminacyclopentane from a monomeric Al^I complex[19]. The proposed mechanism is consistent with simultaneous formation of **10–12**, as well as **10** being the kinetically preferred product (rate determining step 3.3 versus 15.4 kcal mol⁻¹).

To conclude we have reported the first isolation of two cyclotrialumanes, **2^p-tol** and **2^m-xyl**. These unique Al^I compounds have been extensively analysed using detailed experimental and computational techniques, revealing an Al₃ core which retains its structure in solution. DFT analysis reveals a bonding picture between each Al–Al bond which is primarily covalent, enhanced by hyperconjugative donor-acceptor character. Furthermore, the cyclotrialumanes have been shown to react as Al^I fragments (at high temperature) and directly as cyclotrialumanes through a series of reactions with ethylene leading to extraordinary 5- and 7-membered M−C ring systems. The molecules therefore present a unique example of small multi-metallic clusters with discrete reactivity.

The synthesis of the cyclotrialumanes is high yielding and reproducible; it also has the potential to be highly tuneable, with an easy to modify ligand system from which divergent structure and reactivity

can be explored. Crucially, cyclotrialumanes fill a gap in our understanding of neutral Al^I species, which can both *exist* and *react* with hitherto unobserved nuclearity.

## Methods

For further detailed synthetic methods, spectroscopic data and analytical data for all new compounds, as well as SCXRD studies and computational details please see the Supplementary Information.

## Data availability

All processed experimental data generated in this study are provided in the Supplementary Information file. All raw data files are available from the corresponding author upon request. The atomic coordinates generated through computational optimisation are included as a separate Supplementary Data (.txt) file. The X-ray crystallographic coordinates for structures reported in this study have been deposited at the Cambridge Crystallographic Data Centre (CCDC), under deposition numbers 2469911-2469921 and 2503435. These data can be obtained free of charge from The Cambridge Crystallographic Data Centre via www.ccdc.cam.ac.uk/data_request/cif.

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

## Acknowledgements

C.B. thanks the Engineering and Physical Sciences Research Council (grant number EP/Y000129/1) for funding. King's College London Net-Zero centre is thanked for studentship funding (M.T.). Jeremy Cockcroft (UCL) and Jens Najorka (NHM) are thanked for helping us access SCXRD. Thomas Hicks and the CBS NMR Facility are thanked for support running NMR spectroscopy experiments. Alberto Collauto (Imperial College London) is thanked for assistance with EPR spectroscopy.

## Author contributions

I.S. and M.d.V.T. designed and conducted experiments and collected and analysed all data. M.T. conducted the computational analysis. L.S.M. supported SCXRD data analysis. T.K. provided computational support and direction and hosted M.T. for a research visit. C.B. conceived the project, acquired funding, supervised the research and wrote the manuscript with input from all authors.

## Competing interests

The authors declare no competing interests.
