## [Transparent Peer Review file · Nature Communications]

A neutral cyclic aluminium (I) trimer

Corresponding Author: Dr Clare Bakewell

Version 0:

Reviewer comments:

Reviewer #3

(Remarks to the Author)

[Note from the Editor: Reviewer #3 assessed the response given to reviewer #1.]

Bakewell and co-workers provided a revised version of their manuscript that addresses the heterogeneous reviewer reports. The reviewers came to different opinions on the novelty of the work and I hence I would like to emphasize the key findings.

Reports on neutral aluminium(I) compounds are rather rare and most of the astonishing amount of work reported in the last years is based on Roesky $\text{Na}(\text{Cp})_2\text{Al}(\text{I})$ and Schnöckels tetrameric AlCp^* . Only recent contributions by the groups of Bakewell, Liu and Power provided new examples. However, these belong to monomeric and dimeric compounds and neutral aluminium ring systems were missing so far. In their current contribution, the authors close this gap of knowledge and provide a synthetic procedure that provides the respective cyclotrialumane in up to very good yields of 82% from readily available precursors. Notably, low yielding rather demanding syntheses are typically the reality in aluminium(I) chemistry. In addition to the compounds itself, its versatile reactivity that yields the formation of mono-, bi- and trimetallic products is of high interest and value. This will possibly lead to many new interesting reactions in the future.

The reviewers requested clarification in certain points, and the authors have done a good job, thus most of the manuscript reads in its current version much better. The part related to the experiments is conclusive and provides a full picture.

With respect to the computational part, all three reviewers mentioned various concerns, and it appears that things have become more complicated. Given that the authors address with Nature Communications a general and broad audience, I strongly suggest revising this part carefully. The former Figure 4 was quite helpful and while the results of the weightage of the canonical forms by NRT may have been trivial, they are still relevant and should be either in the manuscript or the supporting information. One reviewer mentioned "an apples to oranges kind of comparison that assigns too much meaning to very small variations" and it seems that this has not been addressed properly. Can the authors condense the most important findings and provide a Lewis-like representation in the main text and move the detailed but relevant discussion to the supporting information? That would help understanding the key points and provides enough background for those interested in details. It is highly appreciated that the authors could compute the entire potential energy surface but interpretation of Figure 9, having various insets and colours, is hard. Why not splitting Figure 9, in one part for the formation of **compd 9** and the second part for the subsequent reaction toward **12**?

Supporting information:

- Please add the level of theory to all figures showing computational results.
- The table of contents is not consistent with the actual titles of the sections
- Certain figures are of poor quality – please revise
- Please provide an overlay of the experimental and computational UV-vis spectra

Overall, I strongly support publication in Nature Communications after a thorough revision of the computational part to increase the appeal for a broad audience.

Reviewer #4

(Remarks to the Author)

[Note from the Editor: Reviewer #4 assessed the response given to reviewer #2.]

The results given in this paper seem to be valid but in my opinion they are not sufficiently distinct and impactful to warrant publication in Nature Comm. After all the aluminium monomer, dimer and tetramer are already known so it is not highly surprising that the trimer can be made as well by suitable manipulation of the ligand. The reactivity has interesting features but I don't think they are sufficiently new or surprising and similar reactions and products are known for gallium derivatives.

Response to reviewer #3

Bakewell and co-workers provided a revised version of their manuscript that addresses the heterogeneous reviewer reports. The reviewers came to different opinions on the novelty of the work and I hence I would like to emphasize the key findings.

Reports on neutral aluminium(I) compounds are rather rare and most of the astonishing amount of work reported in the last years is based on Roesky $\text{Na}^+\text{Al}(\text{I})$ and Schnöckels tetrameric AlCp^* . Only recent contributions by the groups of Bakewell, Liu and Power provided new examples. However, these belong to monomeric and dimeric compounds and neutral aluminium ring systems were missing so far. In their current contribution, the authors close this gap of knowledge and provide a synthetic procedure that provides the respective cyclotrialumane in up to very good yields of 82% from readily available precursors. Notably, low yielding rather demanding syntheses are typically the reality in aluminium(I) chemistry. In addition to the compounds itself, its versatile reactivity that yields the formation of mono-, bi- and trimetallic products is of high interest and value. This will possibly lead to many new interesting reactions in the future.

The reviewers requested clarification in certain points, and the authors have done a good job, thus most of the manuscript reads in its current version much better. The part related to the experiments is conclusive and provides a full picture.

With respect to the computational part, all three reviewers mentioned various concerns, and it appears that things have become more complicated. Given that the authors address with Nature Communications a general and broad audience, I strongly suggest revising this part carefully. The former Figure 4 was quite helpful and while the results of the weightage of the canonical forms by NRT may have been trivial, they are still relevant and should be either in the manuscript or the supporting information. One reviewer mentioned “an apples to oranges kind of comparison that assigns too much meaning to very small variations” and it seems that this has not been addressed properly. Can the authors condense the most important findings and provide a Lewis-like representation in the main text and move the detailed but relevant discussion to the supporting information? That would help understanding the key points and provides enough background for those interested in details.

We appreciate the reviewer's feedback on this more general point made by the original reviewer in their first assessment of our manuscript. The comment referred to the overall analysis of our combined data on the electronic structure of complex **2**, and we have carefully addressed all issues relating to the electronic structure by taking the valuable comments from this reviewer into account. Specifically, the interpretation of the Electron Localisation Function has been revised and carefully reworded. Along with this, inconsistent language and interpretation has also been corrected within the NBO and QTAIM sections. The largest concern related to the "asymmetry" in the Al_3 triangle, which we have now shown to be a result of the steric congestion of the co-ligand which induces slight distortions away from local C_3 symmetry. The analysis of the electronic structure and additional results (added to the ESI) complements this observation, and any ambiguity in our initial interpretation has been removed. We believe that this fully addresses the above issue. We fully

stand over our revised interpretation of the data, which is now fully aligned with the suggestions made by the initial reviewer.

Having revisited the manuscript with the reviewers' comments in mind we fully appreciate that the DFT analysis of the Al₃ atomic structure had become longwinded and difficult to interpret. As suggested, we have now removed large sections of the more detailed DFT analysis and redistributed it to the SI. We have also endeavoured to simplify and clarify some of our original discussion, which we hopes makes the interpretation of the data clearer. We have included a revised version of the old Figure 4 in the supporting information but have chosen not to include the dissociation of the trimer in the main text as it seemed to convey a confusing message about the reactivity.

It is highly appreciated that the authors could compute the entire potential energy surface but interpretation of Figure 9, having various insets and colours, is hard. Why not splitting Figure 9, in one part for the formation of cmpd 9 and the second part for the subsequent reaction toward 12?

As suggested, we have split this figure in two to improve readability.

Supporting information:

- Please add the level of theory to all figures showing computational results.

This information has been provided in either figure captions or in subsection titles for all computational data.

- The table of contents is not consistent with the actual titles of the sections

This has been amended.

- Certain figures are of poor quality – please revise

Higher quality figures have been provided.

- Please provide an overlay of the experimental and computational UV-vis spectra

This has been added (Figure S41)

Overall, I strongly support publication in Nature Communications after a thorough revision of the computational part to increase the appeal for a broad audience.

We thank the reviewer for their support, encouraging words and useful suggestions – it is all gratefully received.